# Characterization of the anti-pathogenic, genomic and phenotypic properties of a *Lacticaseibacillus rhamnosus* VHProbi M14 isolate

**Jingyan Zhang**, **Kailing Li**, **Xinping Bu**, **Shumin Cheng**, **Zhi Duan** *

Qingdao Vland Biotech Inc, Nutrition & Health Technology Center, Qingdao, China

* duanzhi@vlandgroup.com

**Data Availability Statement:** All relevant data are within the paper and its Supporting information files.

## Abstract

A strain of lactic acid bacteria from cheese was isolated, that showed strong growth inhibitory effects on *Streptococcus mutans*. The API 50CH system and 16S rDNA sequencing verified that this was a novel strain, and was named *Lacticaseibacillus rhamnosus* VHProbi M14. The strain inhibited the growth of *S. mutans* and *Fusobacterium nucleatum* under mixed culture conditions, coaggregated with *S. mutans* and *F. nucleatum*, and reduced the adhesion of *S. mutans* and *F. nucleatum* on cultured human primary gingival epithelial (HPGE) cells. The pH, peroxidase and protease sensitivity testing found antibacterial substances of protein- and peptide-like structures in addition to organic acids. The antimicrobial substances were sensitive to hydrolysis with trypsin, papain and pineapple protease and were inactived at temperatures above 100˚C. Ammonium sulphate-precipitated proteins from the M14 strain retained the ability to inhibit the growth of *S. mutans* and *F. nucleatum*. The M14 strain contained 23 bacteriocin-related genes encoding for metabolites, belonging to class II bacteriocins. The M14 strain also showed inhibitory effects on 8 other pathogenic strains (*A. actinomycetemcomitans*, *C. albicans*, *E. coli*, *G. vaginalis*, *P. acnes*, *P. gingivalis*, *S. aureus*, *S. enteritids*), and thus has a broad spectrum of bacterial inhibition. This new isolate has been identified as having potential to be used as a probiotic bacterium in clinical applications.

## Introduction

Oral diseases can be acute or chronic with a high global prevalence, and oral health is a primary component of in the maintenance of human health. Oral disease is associated with a wide variety of illnesses and disorders, including dental caries, periodontal disease, tooth loss, oral cancer [1], resulting from oral infections of some pathogenic bacteria, such as *Streptococcus mutans* [2, 3] and *Fusobacterium nucleatum* [4, 5]. A healthy oral cavity can be maintained by promoting advantageous bacteria that inhibit those bacterial species causing oral diseases [6].

Lactic acid bacteria (LAB) constitute an order of gram-positive bacteria sharing common morphological, molecular, and physiological characteristics. They are nonsporulating, non-

**Funding:** This study was supported by the grants from the Mountain Tai New Strategy Industry Leader Program (Grant No. tscy20180317). The person who received the financial support was Zhi Duan.

**Competing interests:** The authors have declared that no competing interests exist.

respiring, but have aerotolerant cocci or rods, and produce lactic acid as one of the main products of carbohydrate fermentation [7]. LAB are present in the mouth, intestines, skin, and vagina of humans [8, 9], and may account for approximately 1% of the cultivable oral microbiota [10]. The most commonly isolated Lactobacilli in the oral cavity are *L. casei*, *L. delbrueckii*, *L. fermentum*, *L. rhamnosus*, and *L. salivarius* species [11–15]. The oral cavity also contains a number of other genera, such as *S. mutans*, *Porphyromonas gingivalis*, *F. nucleatum*, and *Prevotella intermedia* [16]. These bacteria can be either pathogenic or non-pathogenic to the host, that concomitantly inhabit the oral cavity. However, with over-proliferate, microbial dysbiosis can lead to the development of a variety of oral diseases such as dental caries and periodontitis [17]. The main cause of dental caries is that *S. mutans* forms plaque biofilms through surface protein binding to sites on acquired membranes on the tooth surface. The inhibition and removal of dental plaque biofilm are thus important for preventing caries [18]. Probiotics can disrupt the biofilms formed by *S. mutans* by co-agglutinating and adhering to pathogenic bacteria [18, 19]. Recently, the antimicrobial properties of LAB and their roles in the prevention of common oral diseases have been recognized [20], with the list of the LAB species with probiotic activity rapidly increasing. Some studies have shown the beneficial effects of these probiotics in the oral cavity leading to a reduction in the risk of dental caries. For example, Zhang et al reported that *L. plantarum* K41 effectively inhibited *S. mutans*-caused biofilm formation and thus possesses a potential inhibitory effect on dental caries in vivo [21]. Nase et al. (2001) reported reduction of early childhood caries (ECC) in children consuming *L. rhamnosus* GG fermented milk products containing antimutans immunoglobulin G [22]. Simark-Mattsson et al. (2007) showed that *L. paracasei*, *L. plantarum* and *L. rhamnosus* were strong inhibitors both reference and clinical isolates of *S. mutans* [14]. Nikawa et al. (2004) showed that consumption of bovine milk fermented with *L. reuteri* was also effective in oral carriage reduction of *S. mutans*, resulting in a reduction in the risk of dental caries [23].

The antimicrobial mechanisms of LAB are complicated with the production of organic acids such as lactic and acetic acids from carbohydrate fermentation [20, 24]. These organic acids can enter the cells of pathogenic bacteria to reduce intracellular pH and affect cell metabolism [24]. Some bacteria such as *Bifidobacterium*, *Lactobacillus*, *Lactococcus*, and *Pediococcus* produce hydrogen peroxide and/or bacteriocins (proteinaceous compounds) that exhibit antimicrobial properties [24–27]. Hydrogen peroxide is a strong oxidising agent that may rapidly penetrate the cell walls of microorganisms and inactivation the cells [28]. Bacteriocins isolated from LAB can destroy the integrity of the outer membrane of pathogenic bacteria [29].

Considering potentials of using LAB in the prevention of oral diseases, this research aimed to screen LAB strains from various sources for their capacity to inhibit oral pathogenic bacteria. We obtained 11 strains that showed significant inhibitory effects on the growth of *S. mutans*, and the isolate with a strong inhibition effect was identified as a novel *Lactobacilli rhamnosus* strain. We nominally named the bacteria *L. rhamnosus* VHProbi M14, and investigated its anti-pathogenic, genomic and phenotypic characteristics.

## Materials and methods

### Isolation and identification of LAB strains

Yoghurt, kimchi, cheese, and soya juice acquired from various retail stores and samples collected further analysis. The samples were immediately placed on ice and transported to Vland Biotech Company Laboratory, Qingdao, China. Samples (10 g) were weighed, diluted by 10 folds with normal saline, and homogenized with a paddle homogenizer (Scientz-11L, Ningbo, China) at 10 strokes/s for 5 min. The homogenate was then sequentially diluted by 10fold 3 times ($10^{-1}$, $10^{-2}$ and $10^{-3}$). Diluted samples, 100 μL each, were cultured on Man-Rogosa-

Sharpe (MRS) agar plates in anaerobic conditions at 37˚C for 48 h. Colonies grew on the plates for 2 days, and those colonies of different shapes were selected for Gram staining and were microscopically examined. The rod-shaped and Gram-positive strains were preliminarily considered to be potential LAB candidates.

Bacteriostatic activity of the isolated potential LAB strains was then assessed using the Oxford cup diffusion agar method with some modifications as follows [21, 30]. *S. mutans* was initially used as the indicator strain. Firstly, 50 μL of *S. mutans* suspension ($10^8$ CFU/mL) was evenly spread with a spreader on brain-heart infusion (BHI) agar medium in Petri dishes. Then, Oxford cups were placed on the surfaces of the cultures, and 100 μL of the isolate strain suspension was dripped onto the surface. Finally, prepared Petri dishes were incubated at 37˚C for 48 h to allow the pathogen to grow. The diameter of the inhibition zone was measured to evaluate the antibacterial activity of the isolates. All tests were performed in triplicate.

The isolates were subjected to a standard biochemical test using the API CH50 system (Biomerieux, Marcy, l'Etoile, France) and 16S rDNA sequence analysis. A large fragment of the 16S rDNA gene was amplified by PCR, using the universal primers 27F [5'-AGAGTTTGA TCCTGGCTCAG] and 1492R [5'- GGTTACCTTGTTACGACTT] [31]. PCR amplification conditions consisted of preheating at 95˚C for 4 min, 30 cycles of 95˚C (40 s), 42˚C (40 s), and 72˚C (2 min), plus one additional cycle with a final 20-min chain elongation [32]. The PCR products were then purified with a Wizard PCR Preps DNA Purification System and sequenced with a Big Dye Terminator Cycle Sequencing Ready Reaction kit (BGI, China) and a model 310 automatic sequencer. The closest known relatives of the new isolates were identified by database sequence searches, and the sequences of closely related strains were retrieved from the GenBank libraries or Ribosomal Database Project databases (https://blast.ncbi.nlm. nih.gov/Blast.cgi). A phylogenetic tree was then constructed using the NJ (Neighbor-Joining) method with MEGA 11.0 software (https://www.megasoftware.net/), and the bacterial species were identified. Those strains identified as LAB (referred to the isolate or isolates in the following context) were then used in further studies.

## Pathogenic bacteria and culture conditions

Pathogenic bacterial strains tested in this study and their culture conditions are shown in Table 1. These are pathogens that can cause oral diseases, skin diseases, intestinal diseases and female gynaecological diseases, and are purchased from the BeNa Culture Collection Centre (Beijing, China).

**Table 1. Pathogenic bacteria and their culture conditions.**

| Pathogenic bacteria | Culture conditions |
|---|---|
| *Aggregatibacter actinomycetemcomitans* BNCC336945 | Grown on Columbia blood agar (CBA) plates for 48–72 h in anaerobic conditions at 37˚C |
| *Fusobacterium nucleatum* BNCC336949 | |
| *Gardnerella vaginalis* BNCC 354890 *Porphyromonas gingivalis* BNCC353909 | |
| *Candida albicans* ATCC10231 | Grown on sabouraud dextrose agar (SDA) plates for 24 h in aerobic conditions at 37˚C |
| *Escherichia coli* ATCC25922 | Grown on nutrient agar (NA) plates for 24 h in aerobic conditions at 37˚C |
| *Salmonella enteritids* ATCC14028 | |
| *Staphylococcus aureus* ATCC 6538 | |
| *Propionibacterium acnes* ATCC6919 | Grown on reinforced clostridial agar (RCA) plates for 24 h in anaerobic conditions at 37˚C |
| *Streptococcus mutans* ATCC25175 | Grown on brain-heart infusion agar (BHIA) plates for 24 h in aerobic conditions at 37˚C |

## Growth inhibition test, coaggregation test, and antagonistic adhesion test with HPGE cells

Two strains of oral pathogens (*S. mutans* and *F. nucleatum*) were used as the indicator bacteria to determine the antibacterial inhibitory properties of the isolated LAB strains.

The isolated LAB strains were scribed from glycerol tubes onto MRS agar plates and incubated anaerobically at 37˚C for 48 h. Single colonies were then picked from the tubes and incubated in MRS broth medium at 37˚C for 24 h. *S. mutans* was cultured in BHI broth medium, and *F. nucleatum* was cultured in BHI broth medium supplemented with 5% bovine serum.

The isolates was tested for inhibitory effect on pathogenic bacteria by co-culture in vitro according to the method described by Yang [33]. The BHI broth (10 mL) was initially mixed with 2.0% (v/v) of the isolated strain culture ($10^9$ CFU/mL) or *S. mutans* ATCC 25175 culture ($10^9$ CFU/mL) as the inoculants, and cultured at 37˚C for 30 h. Samples were taken at 0, 4, 8, 16, 24, and 30 h, respectively, to determine the viable bacterial counts (CFU/mL) for each species. *F. nucleatum* was cultured in an anaerobic condition. The selective media used were respectively: MRS agar for *Lactobacillus*, Mitis Salivarius Bacitracin agar for *S. mutans*, and Differentia Clostridial Agar supplemented with 5% sheep's blood for *F. nucleatum*. The distinction between colonies of *F. nucleatum* and the isolated strain *(Lacticaseibacillus)* was determined with the aid of a microscopic (Axiolab5, Carl Zeiss, Suzhou, China.). All tests were conducted in triplicate. The inhibition rate (%) was calculated as follows:

$$Inhibition\ rate(\%)$$
$$= (1 - Count\ in\ pathogen\ and\ M14\ mixture/Count\ in\ pathogen\ control) \times 100.$$

The isolates were tested for their congregation capacity with *S. mutans* and *F. nucleatum*. The test was performed using a spectrophotometric assay [21, 34] with some modifications. The cells were harvested by centrifugation at 5,000×g for 15 min, washed three times with the coaggregation buffer (pH 7, containing 0.001 Mol/L Tris(hydroxymethyl)aminomethane) 0.1 mmol/L CaCl$_2$, 0.1 mmol/L MgCl$_2$, and 0.15 Mol/L NaCl. Cisar et al. [35], and re-suspended in the coaggregation buffer at a concentration of approximately $10^9$ CFU/mL. The equal volumes (1 mL each) of the isolate, *S. mutans* or *F. nucleatum* suspension and a coaggregation pair (the isolate + *S. mutans* or the isolate + *F. nucleatum*) suspension were mixed, vortexed for 10 s, and incubated at 37˚C for 2, 4, and 6 h precipitate the cell aggregates. The supernatant fluid (0.2 mL) was carefully removed for optical density readings (OD) at 600 nm using a spectrophotometer (Multiskan FC, Thermo scientific Shanghai, China). All tests were conducted in triplicate. The percentage of co-aggregation was calculated as follows [21]:

$$Coaggregation\ rate\ (\%) = [(A_0 + B_0)/2 - C_t]/(A_0 + B_0)/2 \times 100.$$

where $A_0$ and $B_0$ represent the OD values of the isolate, and *S. mutans* or *F. nucleatum* respectively at time 0, and $C_t$ represents the OD values of the mixture 2, 4, 6 h, respectively.

Adhesion of oral bacteria to the surface of the oral cavity is required for colonization and for subsequent development of disease, therefore, prevention of adhesion of pathogens should prevent the disease [36]. In this study, human primary gingival epithelial (HPGE) cells (iCell Company, Shanghai, China) were used to determine the antagonistic adhesion of the isolated bacteria to *S. mutans*, or *F. nucleatum*. The test was performed using the Esteban-Fernández method with some modifications [37]. Adhesion tests were performed onto a 24-well chamber slide plate. The HPGE cells were seeded ($2.5 \times 10^5$ cells/well) and incubated at 37˚C in 5% CO$_2$ for 18 h, then briefly washed twice with the medium before the following testing was undertaken.

The fresh isolated strain and pathogen (*S. mutans* and *F. nucleatum*) cultures ($10^9$CFU/mL) were washed twice with phosphate-buffered saline (PBS, pH 7.0) and suspended with the same volume of Roswell Park Memorial Institute (RPMI) 1640 culture medium that contained 10% fetal bovine serum. The $OD_{600}$ absorbance was adjusted to 0.4–0.5 with the 1640 culture medium. The isolated strain and pathogen suspensions were mixed at a ratio of 1:1, and 500 µL of the mixture was added onto the prepared 24-well HPGE plate. Each treatment had three wells (repeats). HPGE cells incubated with *S. mutans* or *F. nucleatum* alone were used as the controls. The plates were incubated at 37˚C in 5% $CO_2$ for 120 min. The cells were then washed three times with PBS, fixed with methanol, Giemsa stained for 5 min, washed with PBS, and air dried. Pathogenic bacterial adhesion to each HPGE cell was observed using a microscope with a 100× oil lens (Axiolab5, Carl Zeiss, Suzhou, China.). Fifty cells in each well were observed and the pathogens adhered onto the surface of each HPGE cell was counted, and the mean for 50 counts were calculated for each well. The adhesion index was then calculated as followings:

$$\text{Adhesion index} = \text{Count of adhered pathogen}/\text{Number of HPGE cells}\,(\text{i.e., } 50 \text{ cells}).$$

## Antibacterial test against pathogens

The antibacterial effect of the isolated strains on 10 strains of pathogenic bacteria was tested, as shown in Table 1. The isolates were grown in MRS broth at 37˚C overnight. The broth was then centrifuged at 6,500×g for 5 min, and the supernatant was harvested and filtered through a 0.22µm membrane. Both the fermentation broth and supernatant were used to test their antibacterial activity against the pathogenic bacteria. Oxford cup agar diffusion assay [21, 30] as described above was used, except with the *Candida albicans* strain. The inhibition of the isolates of *C. albicans* was tested using the method described by Simark-Mattsson and Riley [14, 38] with some modifications. Firstly, 8 µL of overnight cultures of the isolated strain was spotted onto a MRS Petri dish, and incubated at 37˚C for 48 h to allow some inhibitory compounds to be metabolized. Then, 7 mL of Stachybotrys glucose semi-solid medium containing 0.5% (v/v) of *C. albicans* ($10^8$ CFU/mL) was poured onto the plate. When the plate was solidified, the plate was then incubated anaerobically at 37˚C for 24–48 h. The plate was examined regularly for presence of an inhibition circle around the colonies. All tests were conducted in triplicate.

## Determination of antibacterial substances in the isolates

Many substances, such as organic acids, hydrogen peroxide and bacteriocins, produced by LAB have the ability to inhibit bacteria, so the compounds in supernatant of the cultured isolates were tested for antibacterial effects on *S. mutans* and *F. nucleatum* as followings. All tests were conducted in triplicate.

**Hydrogen peroxide production.**   Hydrogen peroxide from the isolated strain was determined using a modified method described by Eschenbach et al. [39]. The isolate was cultured on MRS plates containing 10 g/L glucose, 0.25 g/L tetramethylbenzidine, and 0.01 g/L horseradish peroxidase (200 U/mg protein, Macklin, Shanghai, China) at 37˚C in a $CO_2$ atmosphere for 48 h. Colonies showing blue-coloured rings around the colony were classified as being $H_2O_2$ producers.

**Heat sensitivity and pH.**   Supernatants from antimicrobial cultures grown in MRS for 48 h was divided into six aliquots. Two aliquots were adjusted with 1 Mol/L of HCl or 1 Mol/L of NaOH to pH 5.5 and 6.8 respectively. Three aliquots were adjusted with 1 Mol/L of NaOH to pH 6.8, and treated, respectively, at 80˚C and 100˚C for 30 min, or 120˚C for 15 min [40]. The Oxford cup agar diffusion test was used to determine antibacterial activity in 100 µL of the supernatant.

**Sensitivity to proteolytic enzymes.** The sensitivity of the antibacterial substance in the supernatant of the isolated strains to proteolytic enzymes was tested using the method described by Hong and Wang [41, 42]. Supernatant was prepared by centrifuging the cultured isolate at 6500×g for 5 min, transferred into a clean test tube, and its pH was adjusted to 6.8 with 0.1 Mol/L sodium hydroxide. Proteolytic enzymes, trypsin (250 U/mg, Sigma, Saint Louis, American), papain (100 U/mg, Doing Higer, Ningjing, China), and pineapple protease (100 U/mg, Doing Higer, Ningjing, China) were respectively dissolved in phosphate buffer (pH 6.8) to make a concentration of 100 mg/mL. 200 uL of the enzyme preparation was then added into 20 mL of supernatant, so the final concentration of the enzyme was 1 mg/mL. The supernatant containing the enzyme was incubated at 37˚C for 30 min and then at 80˚C for 10 min to inactivate the enzyme. Before evaluating the antibacterial activity, the pH value was checked and readjusted to the initial pH at 6.8 with 1 Mol/L of HCl. The Oxford cup agar diffusion method was used to determine antibacterial activity in 100 μL of the supernatant.

**Ammonium sulphate-precipitated proteins [43].** The supernatant was obtained by centrifuging the culture of the isolated strains at 13,000×g for 12 min. Ammonium sulphate was then added into the supernatant to achieve a saturation level of 70%, stirred slowly for 1 h and left to settle at 4˚C overnight to precipitate proteins of the isolated strain. The solution was centrifuged at 13,000×g for 12 min, and the supernatant was discarded. The precipitate was dissolved in the 1/10 of the original volume of phosphate buffer (25 mmol/L, pH 6.8). The Oxford cup agar diffusion method was used to determine the antibacterial activity in 100 μL of the precipitated protein solution.

**Bacteriocin production.** Bacteriocin production of the isolated strains was determined using the method described by Sookkhee et al. [40] with some modifications. Briefly, the culture of the isolates strain in MRS broth (Luqiao, Beijing) under $CO_2$ at 37˚C for 48 h was centrifuged at 6,500×g for 5 min, and supernatant was collected and adjusted pH at 7.0. Supernatant was lyophilized, and the powder was then constituted in distilled water at a concentration of 600 g/L and filtered through a 0.22 μm membrane before testing against *S. mutans* and *F. nucleatum*. In the test against *S. mutans*, the reconstituted supernatant was firstly diluted by 2-fold using doubled BHI broth, which was then further diluted with BHI broth to the final concentrations of 150, 120, 90, 60, 30, 18, and 0 (BHI buffer only) g/L. 200 uL of each solution was added to a 96-well, flat-bottom microtitre plate, with triplicates for each concentration. Then, 2 uL of *S. mutans* suspension ($10^9$ CFU/mL), incubated overnight before being used, was added to each well, and 50 uL of liquid paraffin was then added carefully to the surface on to prevent moisture loss. The plates were then placed in an aerobic chamber and incubated at 37˚C for 30 h. When tested against *F. nucleatum*, the concentrations of the reconstituted supernatant were 18, 15, 12, 9, 6, 3, and 0 g/L, and 5 uL of the *F. nucleatum* suspension ($10^9$ CFU/mL) was added to each well. The $OD_{600}$ value (Multiskan FC, Thermo Scientific Shanghai, China) was recorded in 60 min intervals for 30 h and used to calculate the inhibition rate as follows:

$$\text{Inhibition rate } (\%) = (1 - OD_{600}N/OD_{600}C) \times 100,$$

where $OD_{600}N$ is the $OD_{600}$ value for a given concentration of the reconstituted supernatant, and $OD_{600}C$ is the $OD_{600}$ value for the control (0 g/L).

The inhibition rates and the corresponding supernatant concentrations were plotted to calculate the $IC_{50}$ value using the SPSS program (IBM SPSS Statistics 20 V20.0, America).

## Whole genome sequencing the isolate

The isolates, freshly cultured in broth was inoculated into 500 mL MRS broth medium at 1% (v/v) inoculum and incubated at 37˚C for 22h, then centrifuged at 10,000×g for 10 min. The bacteria were collected, snap frozen in liquid nitrogen, and sent in dry ice to the Majorbio Sequencing Centre (Shanghai, China). DNA was extracted from the bacterial cultures following recommended procedures of the respective manufacturer (DP302, Tiangen, Beijing, China). A combined strategy using the Illumina Hiseq 2500 and PacBio RS II single-molecule real-time (SMRT) sequencing platforms was used to sequence the genome of this strain [44]. The genome sequences were deposited to public depository databases (https://submit.ncbi. nlm.nih.gov/). Bioinformatics analysis of the genome sequence of the strain was undertaken using the software available on the Majorbio website to determine the bacteriocin gene clusters, the composition of bacteriocin synthesis gene clusters and the type of synthesized bacteriocin. Sequence function annotations were made using NR Database (ftp://ftp.ncbi.nlm.nih. gov/blast/db/). Secondary metabolites were analyzed using Antismash 4.0.2 software (https:// dl.secondarymetabolites.org/releases/4.0.2/). Metabolic system analysis included annotation of carbohydrate-active enzymes [45] and analysis of gene clusters for secondary metabolite synthesis [46]. The sequence function was compared using BLAST+ software (ftp://ftp.ncbi.nlm. nih.gov/blast/executables/blast+/2.3.0/).

## Statistical analysis

All data are presented as a mean ± SD. Where applicable, a 2-tailed Student's t-test was used to analyze the differences between treatments (Excel software; Microsoft, Redmond, WA, USA).

## Results

### Identification of *Lactobacilli rhamnosus* VHProbi M14

A total of 346 bacterial strains were isolated from different sources used in this study. Among them, 11 strains were all identified as LAB and showed significant inhibitory effects on the growth of *S. mutans*. The strain isolated from cheese strongly inhibited the growth of *S. mutans* with an inhibition zone of 2.30 ± 0.15 cm, and identified as a novel *Lactobacilli rhamnosus* strain. We nominally called the strain *L. rhamnosus* VHProbi M14 (referred to the M14 strain) and its detailed characteristics are reported here.

The M14 strain was then subject to a range of biochemical tests. The strain could ferment N-acetylglucosamine, aesculin glycerol, amidon, amygdaline, D-arabinose, L-arabinose, L-arabitol, arbulin, cellobiose, dulcitol, fructose, galactose, gentiobiose, gluconate, glucose, inositol, mannose, mannitol, maltose, melezitose, α-methyl-D-glucoside, lactose, rhamnose, ribose, salicin, sorbitol, sucrose, D-tagatose, and trehalose. The strain was defective in fermenting adonitol, D-arabitol, erythritol, D-fucose, L-fucose, glycogen, inulin, 2-ketogluconate, 5-ketogluconate, D-lyxose, α-methyl-D-mannoside, melibiose, β-methyl-D-xyloside, sorbose, turanoseraffinose, xylitol, D-xylose, and L-xylose. The results were uploaded to the API system and the strain was identified as belonging to genus *L. rhamnosus*. Then, the 16S rDNA sequence of the strain was uploaded to the NCBI database and the strain was found to be close to genus *L. rhamnosus* by BLAST comparison. The 16s rDNA sequences of 19 closely related strains were selected, downloaded, and the phylogenetic tree (Fig 1) was constructed by Neighbor-Joining method using MEGA11.0 software. The M14 strain had the highest homology with *L. rhamnosus* strain DM065.

The M14 strain was thus named *L. rhamnosus* VHProbi M14. The 16s rDNA sequence was uploaded to the NCBI databases (https://www.ncbi.nlm.nih.gov/nuccore/OP824792).

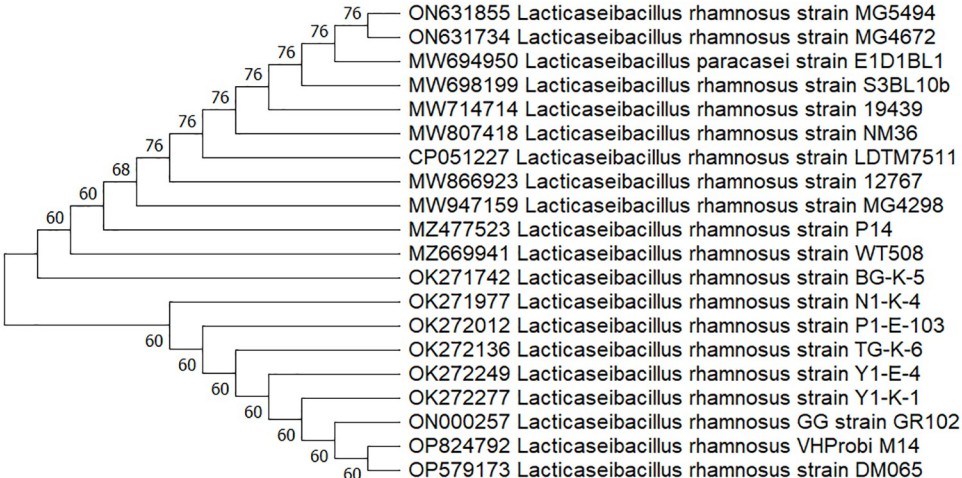

**Fig 1. Phylogenetic tree based on 16S rDNA sequences of *Lacticaseibacillus rhamnosus* VHProbi M14 and related *L. rhamnosus* taxa.**

## Inhibition of the M14 strain on pathogens and antagonistic adhesion of HPGE cells

The effects of the M14 strain on the growth of *S. mutans* and *F. nucleatum* are shown in Fig 2. *S. mutans* in pathogen control started to grow at 4 h and reached the log phase at 8 h. The bacterial counts for *S. mutans* were significantly greater than those for the *S. mutans* + M14 treatment ($P < 0.05$), and the inhibition rate maintained greater than 86% from 8 to 30 h. In the inhibition test for *F. nucleatum*, *F. nucleatum* in pathogen control grew slowly from 4 to 24 h and then reached the log phase, while the bacteria on the *F. nucleatum* + M14 treatment did not show any growth trends. After 16 h, the inhibition rates of the M14 strain on *F. nucleatum* were all greater than 90%.

The M14 strain efficiently coaggregated with *S.mutans* and *F. nucleatum*, are shown in Fig 3. The coaggregation rate of the M14 strain with *S. mutans* and *F. nucleatum* increased over the period of 6 h. Coaggregation rates for strain M14 and *S. mutans* at 2, 4 and 6 h were 31.7% ± 3.1%, 33.8% ± 2.5% and 40.0% ± 0.1% respectively. Co aggregation rates for strain M14 and *F. nucleatum* at 2, 4 and 6 h were 8.1% ± 0.9%, 15.8% ± 2.7% and 21.8% ± 2.4% respectively.

The anti-adhesive effect of the M14 strain on adhesion of *S. mutans* and *F. nucleatum* onto HPGE cells is shown in Fig 4. As shown in Fig 4a1 and 4b1, *S. mutans* and *F. nucleatum* showed a strong attachment on HPGE cells. The adhesion index was 32.8 ± 2.54 for *F. nucleatum* and 10.3 ± 1.52 for *S. mutans* respectively. After a 2-h co-culture of the M14 strain with each of the pathogens, the adhesion index decreased to 18.93 ± 0.69 for *F. nucleatum* and 2.01 ± 0.67 for *S. mutans*, respectively (Fig 4a2 and 4b2). The reductions of the adhesion index were significant ($P < 0.05$). Therefore, the M14 strain showed a significant inhibitory effect on the adhesion of *S. mutans* and *F. nucleatum* onto HPGE cells.

## Antibacterial test of the M14 strain

The antibacterial effects of the M14 strain on the periodontal pathogens (*A. actinomycetemcomitans*, *F. nucleatum*, *P. gingivalis*), the caries pathogen (*S. mutans*), skin pathogens (*P. acnes*, *S. aureus*), the intestinal pathogens (*E. coli*, *S. enteritids*) and vaginitis pathogens (*C. albicans*, *G. vaginalis*) are shown in Table 2. The zone of inhibition of the fementation broth ranged from 1 cm to 2.47 cm, while the zone of inhibition of the supernatant was relatively smaller.

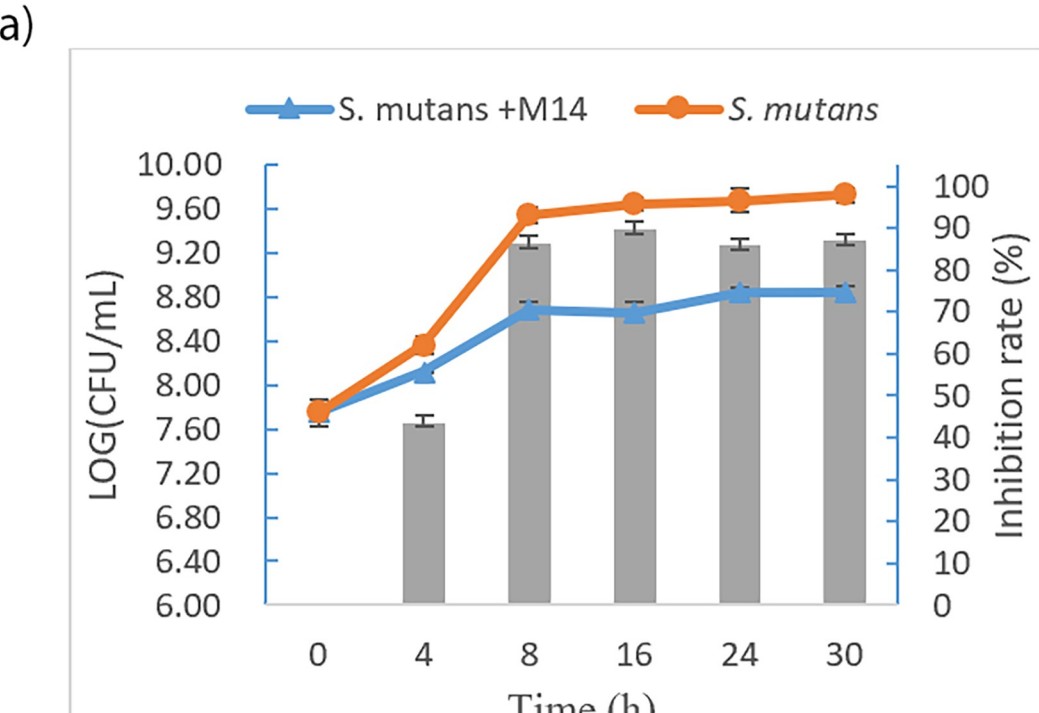

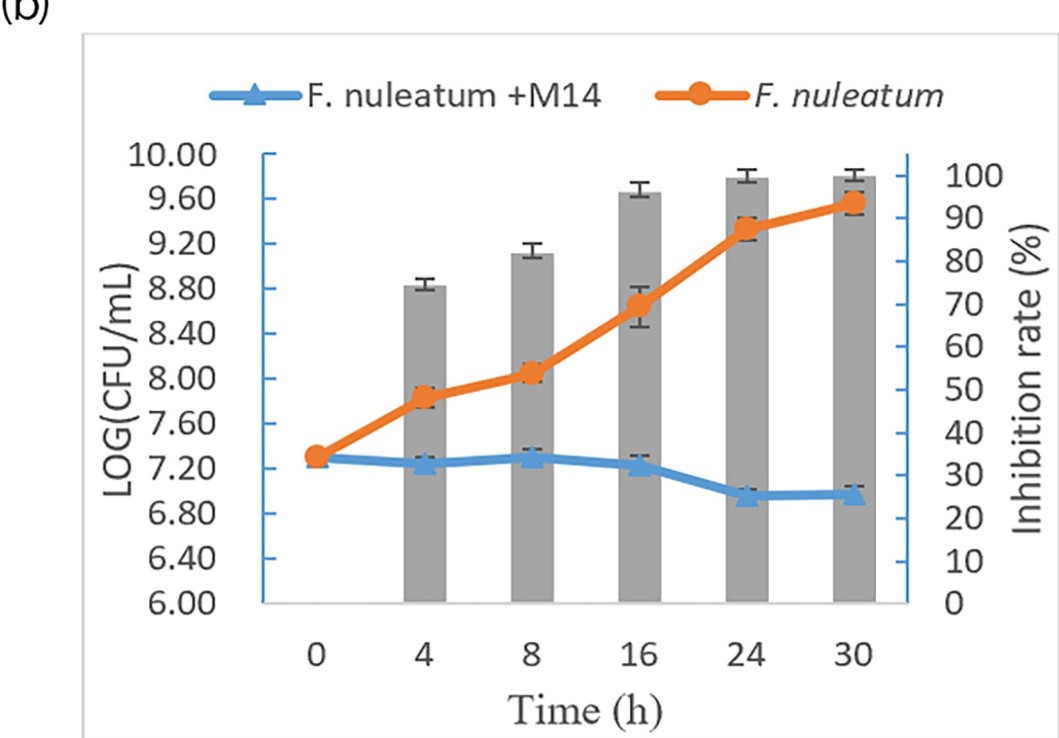

**Fig 2. Growth curves of *S. mutans* and *F. nucleatum* and the inhibition rate (column) of the strain M14 on the growth of *S. mutans* and *F. nucleatum*.** (a) Growth curves of *S. mutans* in mixed culture (*S. mutans* +M14) and control culture; (b) Growth curves of *F. nucleatum* in mixed culture (*F. nucleatum* +M14) and control culture.

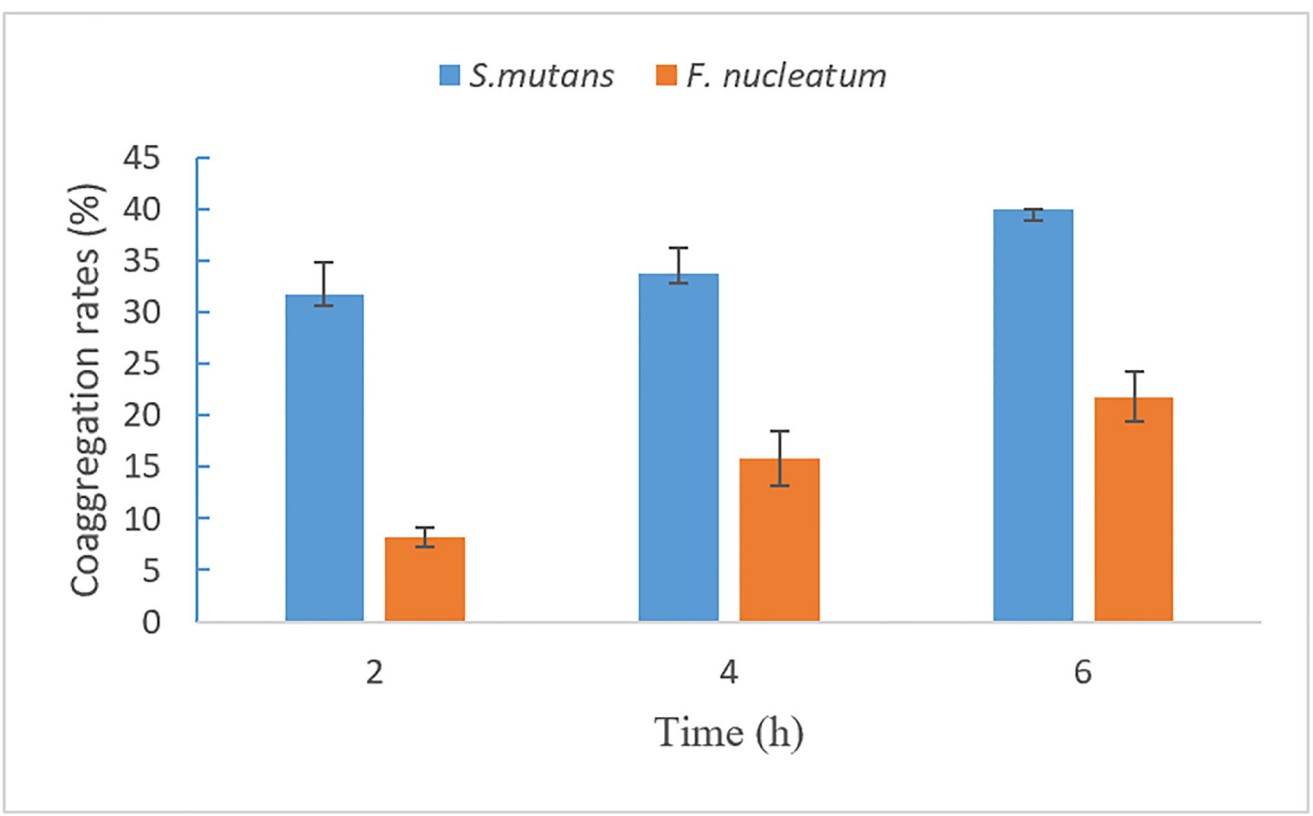

**Fig 3. The co-aggregation rate of *L. rhamnosus* VHProbi M14 with *S. mutans* and *F. nucleatum*.**

The supernatant had no inhibitory effect on *C. albicans* and *P. acnes*, presumably because these two pathogens require higher concentrations of inhibitory substances.

## Antibacterial substances in the M14 strain

The M14 strain did not show any blue-coloured ring around its colonies on plates containing 0.25 g/L tetramethylbenzidine and 0.01 g/L horseradish peroxidase, therefore, the M14 sprain does not produce hydrogen peroxide.

The inhibition capacity of the M14 strain decreased as the pH increased from 4.3 up to 6.8, but remained active at pH 6.8. This indicates the presence of other antibacterial substances in addition to organic acids (Table 3). Compared with the non-enzyme control, the proteolytic treatment of the M14 strain supernatant by trypsin, pineapple protease and papain, reduced the inhibition zone values, indicating that some of the antibacterial substances were proteins or peptides which are sensitive to the proteases. Excluding the interference of organic acids in the supernatant, the zone of inhibition around *S. mutans* and *F. nucleatum* was measured as 1.30 ± 0.05 cm and 1.39 ± 0.01 cm respectively; the inhibition zone values for the supernatant of the M14 strain culture at 80˚C were 1.23 ± 0.03 cm and 1.42 ± 0.06 cm, respectively, indicating that the inhibitory substance was still active, but the antibacterial ability lost fully at temperature 100˚C (Table 3). The ammonium sulphate-precipitated proteins in the M14 strain supernatant also displayed antibacterial activity against *S. mutans* and *F. nucleatum* (Table 3). These tests showed that the antimicrobial substances may consist of organic acids and protein-based antimicrobial peptides.

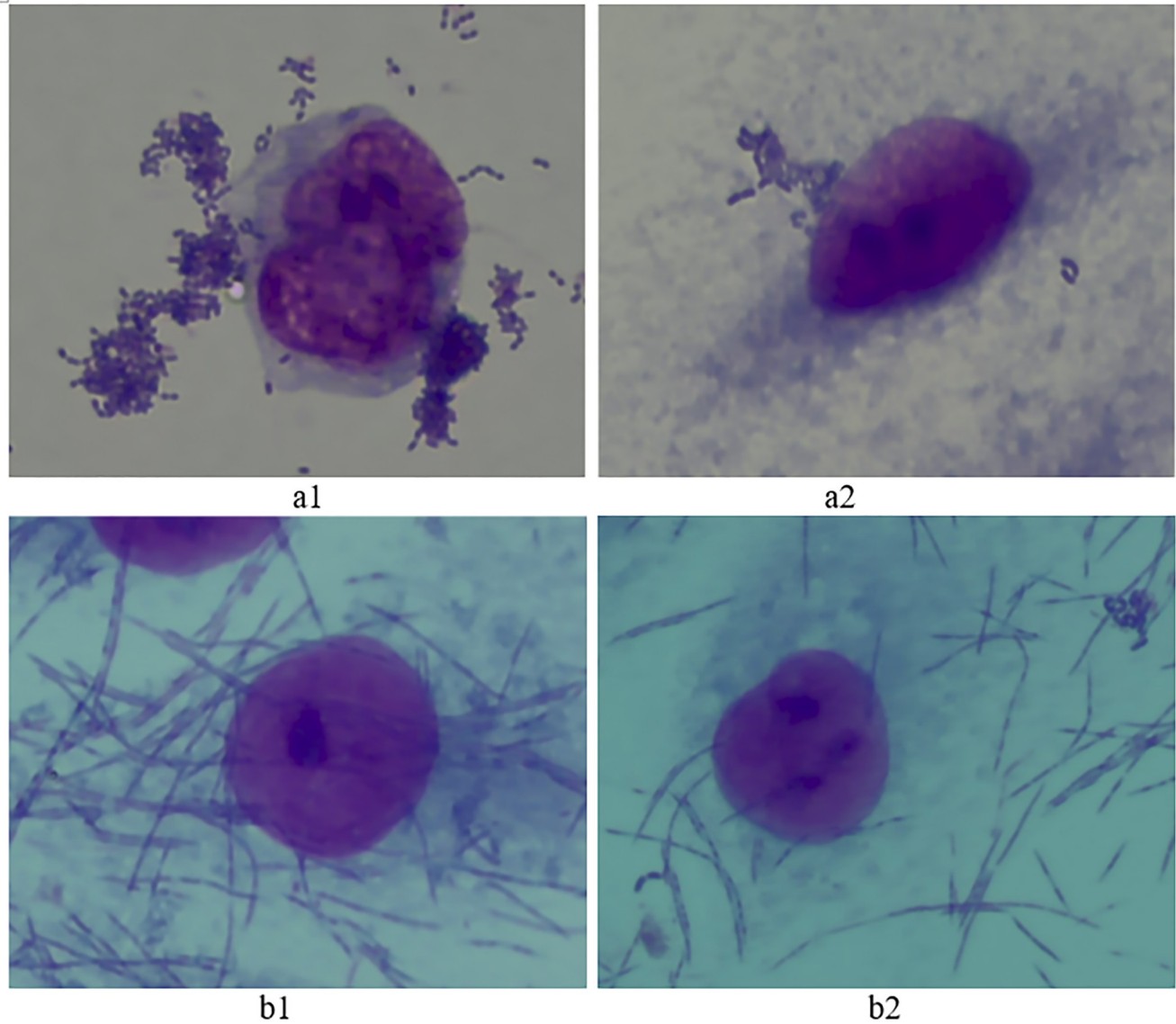

**Fig 4. The effect of *L. rhamnosus* VHProbi M14 on adhesion of *S. mutans* and *F. nucleatum* onto human primary gingival epithelial (HPGE) cells.**
(a1) *S. mutans* control; (a2) M14 + *S. mutans*; (b1) *F. nucleatum* control; (b2) M14 + *F. nucleatum*.

Fig 5 shows the inhibitory effects of the reconstituted lyophilized supernatant of the M14 strain on the growth curves of *S. mutans* and *F. nucleatum*. As the growth period was extended, the reconstituted lyophilized supernatant had significant inhibitory effects on the growth of *S. mutans* (Fig 5a). At the 120 g/L concentration, there was no significant growth of the pathogen in 24 h. Below 120 g/L, the growth of *S. mutans* was inhibited to various degrees, and was depended on the supernatant concentration. *S. mutans* grew more slowly with increasing concentrations of the inhibitory substances. The inhibition rates were calculated separately for different concentrations at 15 hours of the growth (static period) (Table 4). The $IC_{50}$ value was 58.96 g/L for *S. mutans* growth up to 15 h. In the case of *F. nucleatum* (Fig 5), there was insignificant growth of the pathogen within a 30h time period at the 15 g/L concentration. Below concentrations of 15 g/L, the growth of *F. nucleatum* was inhibited to various degrees, and was dependent on supernatant concentrations. *F. nucleatum* grew more slowly with increasing

**Table 2. Antibacterial effects (zone of inhibition, cm) of *L. rhamnosus* VHProbi M14 on pathogens.**

| Pathogenic bacteria | The fermentation broth | The supernatant |
|---|---|---|
| *A. actinomycetemcomitans* BNCC336945 | 1.00 ± 0.10 | 0.93± 0.06 |
| *C. albicans* ATCC10231 | 2.10 ± 0.15 | 0.00 ± 0.00 |
| *E. coli* ATCC25922 | 1.66 ± 0.18 | 1.17 ± 0.08 |
| *F. nucleatum* BNCC336949 | 1.69 ± 0.09 | 1.60 ± 0.05 |
| *G. vaginalis* BNCC 354890 | 1.45 ± 0.15 | 1.18 ± 0.03 |
| *P. acnes* ATCC6919 | 2.47 ± 0.19 | 0.00 ± 0.00 |
| *P. gingivalis* BNCC353909 | 1.40 ± 0.10 | 1.13 ± 0.12 |
| *S. aureus* ATCC 6538 | 1.70 ± 0.20 | 1.47 ± 0.03 |
| *S. enteritids* ATCC14028 | 1.72 ± 0.06 | 1.34 ± 0.11 |
| *S. mutans* ATCC25175 | 2.30 ± 0.15 | 1.70 ± 0.10 |

concentrations of inhibitory substances. The inhibition rates were calculated separately for different concentrations for the first 28 h of growth (static period) (Table 5). The IC$_{50}$ value was 12.52 g/L for *F. nucleatun* for growth in the first 28 h.

### Genome annotation and bioinformatics analysis

The whole genome sequence of the M14 strain was uploaded into the NCBI database (https://www.ncbi.nlm.nih.gov/nuccore/CP095384.1/) and the total length of the genome was 2898403 bp, and could encode 2687 genes, accounting for 84.89% of the whole genome, with an average length of 915.13 bp, with a gene density of 93% and a GC content of 47.29%.

The analysis of gene clusters for secondary metabolite synthesis [43] showed that the secondary metabolites of the M14 strain included mainly SPK3-related coding genes of family I (cluster1) and cluster2 bacteriocin-related coding genes belonging to family II. There were 23 bacteriocin-related coding genes (Fig 6), with gene clusters starting at base 2320116 and ending at base 2339997 on the scaffold. The coding gene numbers ranged from 2201 to 2223, and the predicted functions of these 23 genes is shown in Table 6. Proteins ranged in size from 52 to 479 amino acids and were responsible for a range of biological functions (Table 6).

### Discussion

The present study isolated a novel LAB strain and named it as *L. rhamnosus* VHProbi M14. *S. mutans* and *F. nucleatum* are the main pathogens causing dental caries and periodontal disease

**Table 3. Antibacterial effects (zone of inhibition, cm) of *L. rhamnosus* VHProbi M14 on pathogens under different conditions.**

| Conditions | | *S. mutans* | *F. nucleatum* |
|---|---|---|---|
| **pH** | pH 4.3 (Original) | 1.70 ± 0.10 | 1.60 ± 0.05 |
| | pH 5.5 | 1.42 ± 0.03 | 1.38 ± 0.05 |
| | pH 6.8 | 1.25 ± 0.05 | 1.40 ± 0.02 |
| **Proteases (pH 4.3)** | Non-enzyme control | 1.70 ± 0.10 | 1.65 ± 0.10 |
| | Trypsin | 0.90 ± 0.00 | 1.01 ± 0.03 |
| | Papain | 1.15 ± 0.05 | 1.21 ± 0.02 |
| | Pineapple protease | 1.00 ± 0.00 | 1.20 ± 0.04 |
| **Temperature (pH 6.8)** | Control | 1.30 ± 0.05 | 1.39 ± 0.01 |
| | 80˚C | 1.23 ± 0.03 | 1.42 ± 0.06 |
| | 100˚C | - | - |
| | 120˚C | - | - |
| **Ammonium sulphate precipitation (pH 6.8)** | | 1.30 ± 0.00 | 1.55 ± 0.05 |

(a)

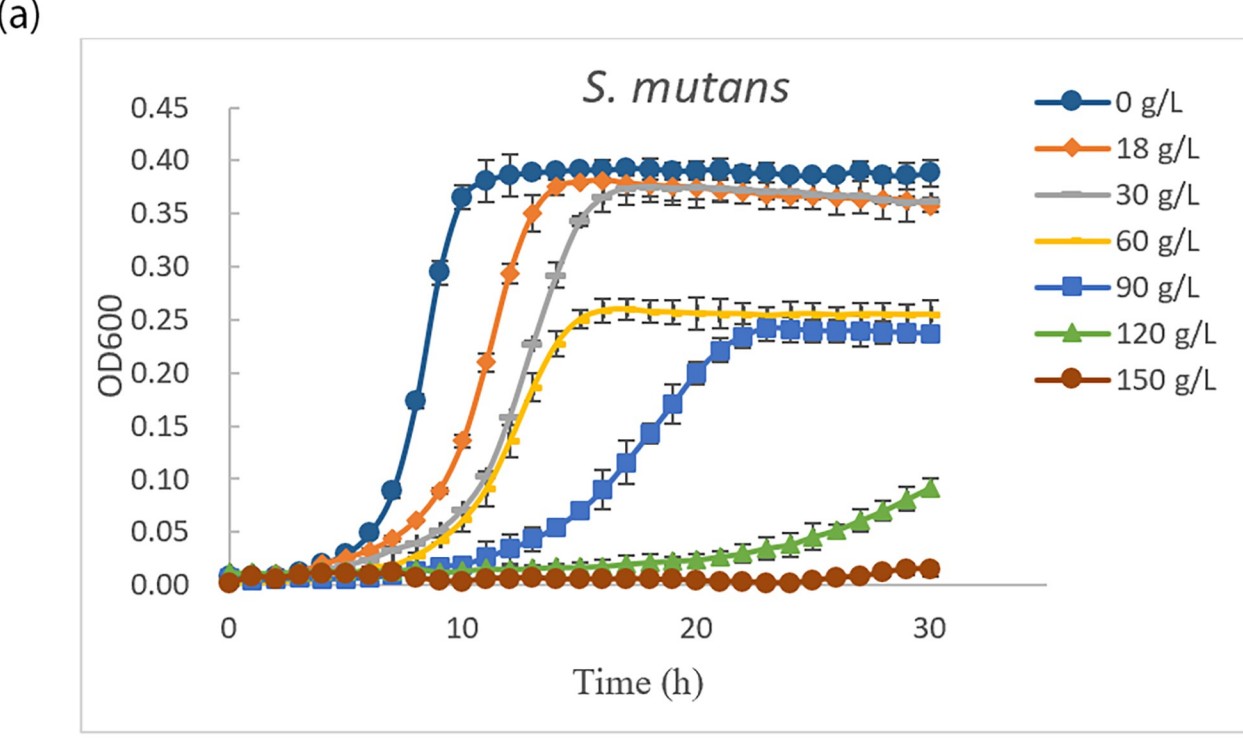

(b)

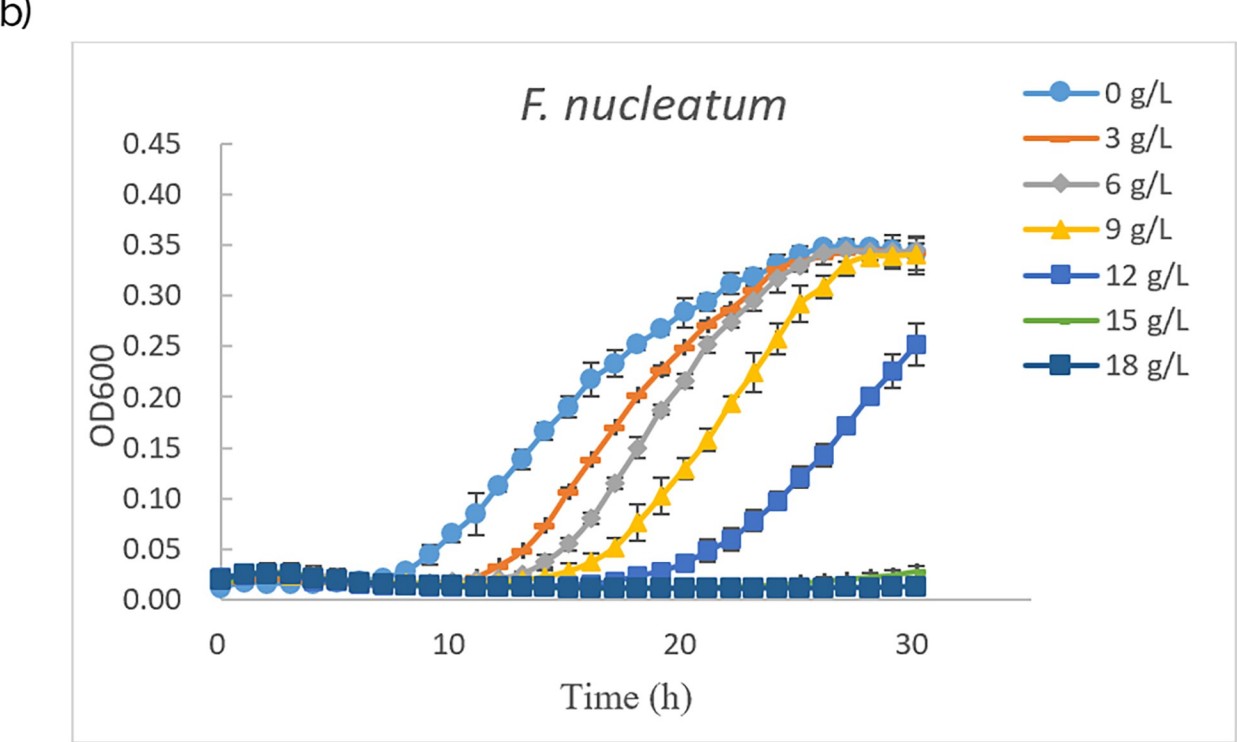

**Fig 5. Antibacterial effects of the concentrations of reconstituted lyophilized supernatant of *L. rhamnosus* VHProbi M14 on the growth of *S. mutans* and *F. nucleatum*.** (a) Growth curves of *S. mutans*; (b) Growth curves of *F. nucleatum*.

**Table 4. The inhibition rates of the concentrations of reconstituted lyophilized supernatant of *L. rhamnosus* VHProbi M14 to the growth of *S. mutans*.**

| Concentrations (g/L) | 0 | 18 | 30 | 60 | 90 | 120 | 150 |
|---|---|---|---|---|---|---|---|
| *S. mutans* | 0.0%±0.0% | 3.1%±0.1% | 12.3%±1.3% | 36.0%±2.2% | 82.2%±1.3% | 95.7%±1.1% | 98.5%±0.2% |

**Table 5. The inhibition rates of the concentrations of reconstituted lyophilized supernatant of *L. rhamnosus* VHProbi M14 to the growth of *F. nucleatum*.**

| Concentrations (g/L) | 0 | 3 | 6 | 9 | 12 | 15 | 18 |
|---|---|---|---|---|---|---|---|
| *F. nucleatum* | 0.0%±0.0% | 0.7%±1.1% | 1.2%±1.2% | 2.5%±0.8% | 41.9%±0.8% | 93.6%±1.2% | 96.6%±1.3% |

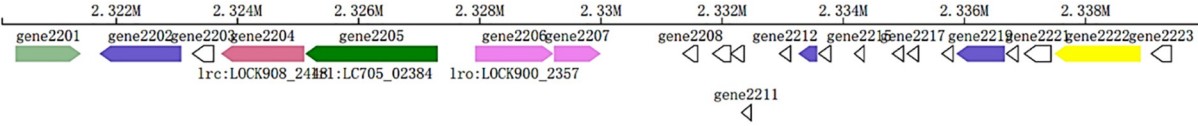

**Fig 6. Bacteriocin synthesis gene cluster mapping of *L. rhamnosus* VHProbi M14.**

in the oral cavity. The M14 strain was found to inhibit the growth of the two pathogens under co-culture conditions, because the M14 strain produced organic acids or other antibacterial substances. The M14 strain efficiently co-aggregated with *S.mutans* and *F. nucleatum*. The levels of aggregation differed between the pathogens and increased with time. Not all *Lactobacillus* strains can aggregate with harmful bacteria [47], so the coaggregation assay may be a useful complement to screen probiotic candidates with possible anti-caries properties. These tests

**Table 6. Details of the gene clusters for the secondary metabolite bacteriocin synthesis by *L. rhamnosus* VHProbi M14.**

| Gene ID | Location | Cluster ID | NR Description | Number of amino acids |
|---|---|---|---|---|
| gene2201 | Chromosome | cluster2 | aldo/keto reductase | 360 |
| gene2202 | Chromosome | cluster2 | threonine/serine exporter family protein | 451 |
| gene2203 | Chromosome | cluster2 | hypothetical protein | 116 |
| gene2204 | Chromosome | cluster2 | bacteriocin secretion accessory protein | 459 |
| gene2205 | Chromosome | cluster2 | peptide cleavage/export ABC transporter | 730 |
| gene2206 | Chromosome | cluster2 | GHKL domain-containing protein | 431 |
| gene2207 | Chromosome | cluster2 | LytTR family transcriptional regulator DNA-binding domain-containing protein | 258 |
| gene2208 | Chromosome | cluster2 | hypothetical protein | 81 |
| gene2209 | Chromosome | cluster2 | bacteriocin immunity protein | 99 |
| gene2210 | Chromosome | cluster2 | - | 73 |
| gene2211 | Chromosome | cluster2 | class IIb bacteriocin, lactobin A/cerein 7B family | 52 |
| gene2212 | Chromosome | cluster2 | MULTISPECIES: bacteriocin | 61 |
| gene2213 | Chromosome | cluster2 | Gar-IM | 105 |
| gene2214 | Chromosome | cluster2 | garvicin Q family class II bacteriocin | 66 |
| gene2215 | Chromosome | cluster2 | MULTISPECIES: class IIb bacteriocin, lactobin A/cerein 7B family | 52 |
| gene2216 | Chromosome | cluster2 | MULTISPECIES: bacteriocin | 61 |
| gene2217 | Chromosome | cluster2 | hypothetical protein | 61 |
| gene2218 | Chromosome | cluster2 | MULTISPECIES: hypothetical protein | 268 |
| gene2219 | Chromosome | cluster2 | CPBP family intramembrane metalloprotease | 268 |
| gene2220 | Chromosome | cluster2 | hypothetical protein | 64 |
| gene2221 | Chromosome | cluster2 | Rrf2 family transcriptional regulator | 146 |
| gene2222 | Chromosome | cluster2 | MFS transporter | 475 |
| gene2223 | Chromosome | cluster2 | hypothetical protein | 110 |

indicate that *L. rhamnosus* VHProbi M14 is a potential oral probiotic. We also found that *L. rhamnosus* VHProbi M14 displayed antibacterial activity on other pathogens, such as oral pathogens *P. gingivalis* and *A. actinomycetemcomitans*, intestinal pathogens *Salmonella* and *Escherichia coli*, skin pathogens *P. acnes* and *S. aureus*, and female vaginal pathogens *G. vaginalis* and *C. albicans*. *Salmonella* and *E. coli* are both pathogens that cause diarrhoea in children [48]. *P. acnes* is a gram-positive bacterium, found ubiquitously as a commensal on the surface of the skin, bowel, conjunctival surface, oral mucosa, and even the external ear canal. It can not only cause acne on the skin, but may also play a role in the pathogenesis of sarcoidosis and ulcerative colitis [49]. *G. vaginalis* is a bacterial vaginitis (BV) pathogen affecting women of childbearing age worldwide [50]. *C. albicans* is the most common fungal pathogen in humans, which can cause a variety of diseases such as periodontitis and fungal vaginitis [51, 52]. The ability of strain M14 to inhibit the growth of these pathogens provides theoretical support for its later application in these areas. We will further research the function in these areas.

We further investigated the characteristics of the antibacterial substances in the M14 strain using *S. mutans* and *F. nucleatum* as indicator pathogen species. The antibacterial substance in the M14 strain was free of hydrogen peroxide but consisted of organic acids and protein-based antibacterial peptides. We identified that bacteriocins (genes 2211, 2214, 2215) belonged to class II bacteriocins. Examples of class II bacteriocins include salivacin 140 from *L. salivarius* [20], and acidocin J1229 from *L. acidophilus* [53], Plantaricin EF, Plantaricin JK from *L. plantarum* L-ZS9 [43]. The activity of bacteriocins of class II is due to pore formation in the cytoplasmic membra [53]. The inhibitory ability of bacteriocins of class IIb is dependent on the synergistic action of the bipeptides [54]. Comparative analysis of the proteins encoded by genes 2211, 2212, 2214, 2215 and 2216 using BLAST showed that these proteins are similar to the bacteriocin proteins of strains *L. rhamnosus* GG, IDCC 3201, HN001 and DSM20021. LGG was able to inhibit the growth of *Escherichia coli*, *Staphylococcus aureus*, *Salmonella paratyphi* B and *Salmonella enterica* [55]. The HN001 strain can also inhibit the growth of methanogens and/or archaea bacteria [56]. Rhamnose IDCC 3201 has potent inhibitory activity against various pathogens responsible for inflammatory responses in the gastrointestinal tract (i.e. *Bacillus cereus*, *Enterococcus faecalis*, *Staphylococcus aureus* and *Salmonella typhi*), respiratory system (i.e. *Streptococcus pneumoniae*) and vagina (i.e. *Candida albicans*) [57]. DSM 20021 strain can reduce the adhesion of pathogenic bacteria [58]. All these strains contain bacteriocin-related genes and have the ability to inhibit the growth of pathogenic bacteria. This also suggests that *L. rhamnosus* VHProbi M14 has the similar genetic background, as these other probiotic bacteria, to synthesize bacteriocins, consistent with M14's antibacterial ability.

Probiotics are also gaining attention as a new type of oral care product. The function of *L. rhamnosus* VHProbi M14 as a potential oral probiotic will be further investigated in animal caries and periodontitis models. We will also later verify the role of this strain in the prevention and treatment of dental caries and periodontitis through clinical trials. We then expect to develop oral care products containing this strain and its metabolites.

## Supporting information

**S1 Table. Antibacterial effects (zone of inhibition, cm) of *L. rhamnosus* VHProbi M14 on pathogens.**
(XLSX)

**S2 Table. Antibacterial effects (zone of inhibition, cm) of *L. rhamnosus* VHProbi M14 on pathogens under different conditions.**
(XLSX)

**S3 Table. The inhibition rates of the concentrations of reconstituted lyophilized supernatant of *L. rhamnosus* VHProbi M14 to the growth of *S. mutans*.**
(XLSX)

**S4 Table. The inhibition rates of the concentrations of reconstituted lyophilized supernatant of *L. rhamnosus* VHProbi M14 to the growth of *F. nucleatum*.**
(XLSX)

**S1 Fig. Growth curves of *S. mutans* and *F. nucleatum* and the inhibition rate (column) of the strain M14 on the growth of *S. mutans* and *F. nucleatum*.**
(XLSX)

**S2 Fig. The coaggregation rate of *L. rhamnosus* VHProbi M14 with *S. mutans* and *F. nucleatum*.**
(XLSX)

**S3 Fig. The effect of *L. rhamnosus* VHProbi M14 on adhesion of *S. mutans* and *F. nucleatum* onto human primary gingival epithelial (HPGE) cells.**
(XLSX)

**S4 Fig. Antibacterial effects of the concentrations of reconstituted lyophilized supernatant of L. rhamnosus VHProbi M14 on the growth of *S. mutans* and *F. nucleatum*.**
(XLSX)

## Author Contributions

**Conceptualization:** Jingyan Zhang, Zhi Duan.

**Data curation:** Jingyan Zhang, Kailing Li, Xinping Bu, Shumin Cheng.

**Formal analysis:** Jingyan Zhang.

**Funding acquisition:** Zhi Duan.

**Investigation:** Zhi Duan.

**Methodology:** Jingyan Zhang, Kailing Li, Xinping Bu, Shumin Cheng, Zhi Duan.

**Project administration:** Jingyan Zhang, Zhi Duan.

**Resources:** Zhi Duan.

**Software:** Jingyan Zhang.

**Supervision:** Jingyan Zhang, Zhi Duan.

**Validation:** Jingyan Zhang, Zhi Duan.

**Visualization:** Jingyan Zhang.

**Writing – original draft:** Jingyan Zhang.

**Writing – review & editing:** Jingyan Zhang, Kailing Li, Xinping Bu, Shumin Cheng, Zhi Duan.

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
