## [Decision Letter · Decision Letter 0]

8 Mar 2023

PONE-D-23-04919Characterization of the anti-pathogenic, genomic and phenotypic properties of a Lacticaseibacillus rhamnosus VHProbi M14 isolatePLOS ONE

Dear Dr. Duan,

Thank you for submitting your manuscript to PLOS ONE. After careful consideration, we feel that it has merit but does not fully meet PLOS ONE’s publication criteria as it currently stands. Therefore, we invite you to submit a revised version of the manuscript that addresses the points raised during the review process.

We look forward to receiving your revised manuscript.

Kind regards,

José António Baptista Machado Soares, PhD

Academic Editor

PLOS ONE

Journal Requirements:

Reviewers' comments:

Reviewer's Responses to Questions

**Comments to the Author**

1. Is the manuscript technically sound, and do the data support the conclusions?

Reviewer #1: Partly

Reviewer #2: Yes

Reviewer #3: Yes

Reviewer #4: Yes

2. Has the statistical analysis been performed appropriately and rigorously? 

Reviewer #1: Yes

Reviewer #2: Yes

Reviewer #3: Yes

Reviewer #4: Yes

3. Have the authors made all data underlying the findings in their manuscript fully available?

Reviewer #1: No

Reviewer #2: No

Reviewer #3: Yes

Reviewer #4: Yes

4. Is the manuscript presented in an intelligible fashion and written in standard English?

Reviewer #1: No

Reviewer #2: Yes

Reviewer #3: No

Reviewer #4: Yes

5. Review Comments to the Author

Reviewer #1: The manuscript entitled “Characterization of the anti-pathogenic, genomic and phenotypic properties of a Lacticaseibacillus rhamnosus VHProbi M14 isolate” is about to provide the characteristics of VHProbi M14 isolate.

At abstract line 10, authors specify the effects of lactic acid bacteria. For example “strong inhibitory effects” should be “strong growth inhibitory effects”.

Line19, It is not clear that the secondary metabolites is belong to bacteriocins? Bacteriocins is not a secondary metabolite. So authors should re-write that sentence.

“The M14 strain contained 23 bacteriocin-related genes encoding for the secondary metabolites, belonging to class II bacteriocins.”

Why authors mention of “bacteriostatic activity” in Line 91? It is not clear the function of bacterial suspension.

It is not clear what is the advantage of this strain compared with previously identified LAB, because the test for antibacterial activity is not referenced and compared with positive control.

Moreover, it is not sure the “Bacteriostatic activity test” I really to check the static or cidal activity.

And, What is the difference of this LAB compare with other LAB?

There are nots of unclear sentence for example “In the test against S. mutans”, “When tested against F. nucleatum,” and so on, and mistakenly written expression for example “point at line 60, OD600, CO2, IC50, membra, and sometimes missed “and”.

Strain name should be abbreviated after first time use.

References are not written consistently.

At figure 1, What is the inhibition rate by pathogenic bacteria only control mean?

Reviewer #2: In this study, the authors set out to characterize the anti-pathogenic, genomic and phenotypic properties of a novel and potential probiotic LAB. The abstract was well written, concisely overviewing the problem statement, methods, results, and conclusion. The study appears well conducted and contributes valuable data to what is already known. Before recommending it for publication, I have the following observations:

1. It is not clear how many repeated experiments (replicates) were done. Authors should include this vital piece of information.

2. For the Growth inhibition step (see line 110), authors should cite the followed method.

3. On what basis did the authors select the pathogens used in this study? Can more information about this be added to the introduction (and, if possible, the discussion) section?

4. Line 247, please change ‘346 strains of bacteria’ to ‘346 bacteria strains’

5. Given the volume of results obtained, the discussion section can be further improved, e.g. G. vaginalis is an economic pathogen affecting women of childbearing age worldwide. Authors should discuss their findings, presenting their novel strain as an alternative to treating infections caused by this pathogen.

6. The manuscript would benefit greatly if a native English speaker could proofread it for further clarity.

Reviewer #3: This manuscript describes the results of an investigation of a novel identified Lactobacillus strain with possible antimicrobial effects on various pathogens.

The topic is of interest to the scientific community as the use of probiotic strains is increasing. However a few clarifications should be made before possible publication of the manuscript.

1. Introduction: Please provide a more sufficient justification of the proposed use of the new LB strain as probiotic organism. The main problem of many infections, especially in the oral cavity, is the occurence of pathogenic biofilms. Please indicate a possible way of action of the new LB strain against established biofilms containing pathogens, as a simple application on mature biofilms will only have very little effect on them.

2. p.2, l.54: ... reported a childhood reduction of dental caries...: what is meant by this phrase? Probably you mean the reduction of eary childhood caries (ECC) in patients consuming the mentioned LB strain.

This is one of some parts of the manuscript which need a thourough revise of the language to provide a more easily readable manuscript. Further, some passages of the Materials and Methods section are quite long, please shorten this section of the manuscript and avoid repetitions.

3. Table 1: Why did you choose aerobic growth conditions for the P. acnes strain? The ATCC recommendation is an anaerobic culture for this strain (ATCC 6919, Cutibacterium acnes), please double check the correct nomenclature of all used microorganisms as these names change often.

4. Discussion: This section needs major revisions as it is mainly a repetition of the results. Please avoid any unnecessary repetitions. Some ideas for improving this section may include a discussion of possible clinical protocols for the use of the new LB strain, as well as some suggestions what further research should be carried out before an in-vivo application is possible.

Reviewer #4: The manuscript “Characterization of the anti-pathogenic, genomic and phenotypic properties of a Lacticaseibacillus rhamnosus VHProbi M14 isolate” is well written, and the study is generally presented with a good design for drawing the conclusions made by the authors. The author isolated and identified a novel LAB strain from cheese that showed the capability to inhibit a series of human pathogens including Streptococcus mutans and Fusobacterium nucleatum. The study brings new results for the LAB characterization that supports earlier studies related to the exploration of clinical probiotics. Still, the manuscript could benefit from some language polishing. I have some minor comments for changes that may be considered during a revision process. If more information is provided as described below, I recommend that the manuscript can be published in Plos One.

Introduction

Line 44: “and produce lactic acid ….”

Line 47-51: The information on other genera is irrelevant to the main context as you did not investigate these bacteria and their interbacterial relationship in the current study.

Line 60- 61: Reference.

Line 67: “in the prevention of oral diseases…”

Line 69: “a strong inhibition effect” or “the strongest inhibition effect”

M&M

The research question is well defined, relevant, and meaningful.

Line 85: where were the S. mautans strains from? Obtained from commercial providers or isolated by the authors?

Line 97: The provider of the “Big Dye TM Terminator Cycle Sequencing Ready Reaction kit”

Line 104-108: It’s good to see you have tested a wide range of pathogens that are related to various diseases. However, it would be better to provide more background information in the section of Introduction on these target pathogens.

Line 111: again, where did you get the strain of F. nucleatum?

Line 164: the reference for Oxford cup agar diffusion assay

Results

Line 247-249: what were those isolated LAB? Why did not show the result?

Line 259-260: How close? What was the similarity?

Line 271-272: “because….” This sentence should be move to the discussion section.

Line 283: reductions of the adhesion index were significant

Discussion

Line 375: these bacteriocin-related genes were…

Line 376: Did you isolate these L. rhamnosus strains?

6. PLOS authors have the option to publish the peer review history of their article (what does this mean?). If published, this will include your full peer review and any attached files.

Reviewer #1: No

Reviewer #2: **Yes: **Smith Etareri Evivie

Reviewer #3: No

Reviewer #4: No

---

## [Author Response · Author response to Decision Letter 0]

15 Mar 2023

reposond to editor:

I have uploaded all the data as Supporting Information files.

Reposond to reviewer1：

1. Line 10, The manuscript had been revised.

2. Line 19, The manuscript had been revised.

3. Line 91, This refers to the isolates with a viable bacterial inhibitor. I have removed this expression as it may be ambiguous.

4. We screened over 300 strains from raw materials, and then screened 11 strains that could inhibit the growth of S. mutans through the Oxford Cup test . The Oxford cup inhibition test is a semi-quantitative assay. We repeated the experiment several times. We found that strain M14 had a more stable ability to inhibit the growth of S. mutans.

5. A number of inaccurate statements in the manuscript had been revised.

6. Fig 2.? In the control group, only pathogenic bacteria grew; in the experimental group, pathogenic bacteria and M14 grew together. The inhibition rate (%) was calculated as follows:

Inhibition rate (%) = (1 – Count in pathogen and M14 mixture / Count in pathogen control) ×100. Inhibition rate refers to the ability of lactic acid bacteria to inhibit the growth of pathogenic bacteria.

Respond to reviewer2:

1. All tests were conducted in triplicate. There were at least three replicates on three different occasions. The manuscript had been revised.

2. The manuscript had been revised.

3. These pathogens were chosen because of the potential application of our strains in this area and I will add some of this section to the discussion.

4. The manuscript had been revised.

5. The manuscript had been revised.

6. I will try to improve my English writing and have it revised by professionals.

Respond to reviewer3:

1. The introduction had been added and revised. The biofilm removal test and the adhesion inhibition test were also done. The biofilm test was compared by counting the number of pathogenic bacteria on the chamber slide and since the thickness of the biofilm was not measured, this test was not added to the article here.

2. Yes, the manuscript had been revised.

3. Propionibacterium acnes was cultured anaerobically. The manuscript had been revised.

4. The discussion section has been stripped of duplication and some new content has been added.

Respond to reviewer4:

1. Introduction. The manuscript had been revised.

2. M&M

 Line 85: These pathogens are purchased from the BeNa Culture Collection Centre (Beijing, China) (Line 113). 

Line 97: The manuscript had been revised.

Line 104-108: I will add the introduction of these pathogens in discussion.

Line 111: These pathogens are purchased from the BeNa Culture Collection Centre (Beijing, China) (Line 113). 

Line 164: The manuscript had been revised.

3. Results

Line 247-249: The isolates were first observed by colony morphology and microscopy as rod-shaped bacteria. Since the main focus of this paper was on strain M14, information on other strains has been omitted.

Line 259-260: the 16S rDNA sequence of the strain was uploaded to the NCBI database and the strain was found to be close to genus L. rhamnosus by BLAST. Similar sequences refer to 16s rDNA.

Line 271-272: The manuscript had been revised.

Line 283: The manuscript had been revised.

4. Discussion

Line 375: The manuscript had been revised.

These strains are from other companies. By BLAST, they had similar bacteriocin related genes with strain M14.

---

## [Decision Letter · Decision Letter 1]

27 Mar 2023

PONE-D-23-04919R1Characterization of the anti-pathogenic, genomic and phenotypic properties of a Lacticaseibacillus rhamnosus VHProbi M14 isolatePLOS ONE

Dear Dr. Duan,

Thank you for submitting your manuscript to PLOS ONE. After careful consideration, we feel that it has merit but does not fully meet PLOS ONE’s publication criteria as it currently stands. Therefore, we invite you to submit a revised version of the manuscript that addresses the points raised during the review process.

Kind regards,

José António Baptista Machado Soares, PhD

Academic Editor

PLOS ONE

Additional Editor Comments:

Dear authors,

Congratulations for the revised manuscript. Please address the minor comments added by Reviewer 1, as follow:

Please add SD in the figure and clarify the test of “Bacteriostatic activity test against other pathogens” because it was not realized a static or bactericidal activity.

Reviewers' comments:

Reviewer's Responses to Questions

**Comments to the Author**

1. If the authors have adequately addressed your comments raised in a previous round of review and you feel that this manuscript is now acceptable for publication, you may indicate that here to bypass the “Comments to the Author” section, enter your conflict of interest statement in the “Confidential to Editor” section, and submit your "Accept" recommendation.

Reviewer #1: (No Response)

Reviewer #3: All comments have been addressed

2. Is the manuscript technically sound, and do the data support the conclusions?

Reviewer #1: Yes

Reviewer #3: Yes

3. Has the statistical analysis been performed appropriately and rigorously? 

Reviewer #1: Yes

Reviewer #3: Yes

4. Have the authors made all data underlying the findings in their manuscript fully available?

Reviewer #1: Yes

Reviewer #3: Yes

5. Is the manuscript presented in an intelligible fashion and written in standard English?

Reviewer #1: Yes

Reviewer #3: Yes

6. Review Comments to the Author

Reviewer #1: I still wonder because authors said “Bacteriostatic activity test against other pathogens”, but that is just to test antibacterial effects not for whether static or cidal activity.

So, authors should change that.

Authors need to add SD in the figure.

Reviewer #3: (No Response)

7. PLOS authors have the option to publish the peer review history of their article (what does this mean?). If published, this will include your full peer review and any attached files.

Reviewer #1: No

Reviewer #3: No

---

## [Author Response · Author response to Decision Letter 1]

29 Mar 2023

Respond to reviewer #1

In the manuscript I have replaced the expression" Bacteriostatic activity test” with "antibacterial test". The other pathogens were only tested in the Oxford Cup.

I have added SD in the figures.

---

## [Editor Report · Decision Letter 2]

6 Apr 2023

PONE-D-23-04919R2Characterization of the anti-pathogenic, genomic and phenotypic properties of a Lacticaseibacillus rhamnosus VHProbi M14 isolatePLOS ONE

Dear Dr. Duan,

Thank you for submitting your manuscript to PLOS ONE. After careful consideration, we feel that it has merit but does not fully meet PLOS ONE’s publication criteria as it currently stands. Therefore, we invite you to submit a revised version of the manuscript that addresses the points raised during the review process.

We look forward to receiving your revised manuscript.

Kind regards,

José António Baptista Machado Soares, PhD

Academic Editor

PLOS ONE

Journal Requirements:

Additional Editor Comments: 

Dear authors,

Please address the minor commentors of the reviewer that I also agree, more exactly: "I still wonder because authors said “Bacteriostatic activity test against other pathogens”, but that is just to test antibacterial effects not for whether static or bactercidal activity. So, authors should change that. Authors need to add SD in the figure."

I believe that after this minor change, that the next version will be suitable for publication.

Thank you and best regards,

António
---

## [Author Response · Author response to Decision Letter 2]

12 Apr 2023

This supernatant inhibition test has been done before for part pathogens. E. coli, P. acnes and S. enteritids didn’t test. I have redone the inhibition tests on these three pathogens and have added the latest test data to the manuscript. The zone of inhibition in the fermentation broth ranged from 1 cm to 2.47 cm, while the zone of inhibition in the supernatant （static or bactercidal activity）was relatively smaller. Because the bacteria in the fermentation broth are still growing and producing antibacterial substances (organic acids or bacteriocins) as it spreads.

The supernatant has no antibacterial activity against C. albicans and P. acnes, presumably because these two pathogens require higher concentrations of inhibitory substances.

 I also repeated the heat sensitivity test for the inhibiting substance (excluding the interference of organic acids) and revised the manuscript.

 I have added SD to the figures.

 I have also revised other errors in the manuscript.

 I have added a legend/caption for figures 1, 2, 3, 4, 5 and 6 in my main document.

---

## [Editor Report · Decision Letter 3]

25 Apr 2023

Characterization of the anti-pathogenic, genomic and phenotypic properties of a Lacticaseibacillus rhamnosus VHProbi M14 isolate

PONE-D-23-04919R3

Dear Dr. Duan,

We’re pleased to inform you that your manuscript has been judged scientifically suitable for publication and will be formally accepted for publication once it meets all outstanding technical requirements.

Kind regards,

José António Baptista Machado Soares, PhD

Academic Editor

PLOS ONE
---

## [Editor Report · Acceptance letter]

5 May 2023

PONE-D-23-04919R3 

Characterization of the anti-pathogenic, genomic and phenotypic properties of a *Lacticaseibacillus rhamnosus* VHProbi M14 isolate 

Dear Dr. Duan:

I'm pleased to inform you that your manuscript has been deemed suitable for publication in PLOS ONE. Congratulations! Your manuscript is now with our production department. 

Kind regards, 

on behalf of

Dr. José António Baptista Machado Soares 

Academic Editor

PLOS ONE